# Guiding Deep Molecular Optimization
# with Genetic Exploration

**Sungsoo Ahn**    **Junsu Kim**    **Hankook Lee**    **Jinwoo Shin**
Korea Advanced Institute of Science and Technology (KAIST)
{sungsoo.ahn, junsu.kim, hankook.lee, jinwoos}@kaist.ac.kr

## Abstract

De novo molecular design attempts to search over the chemical space for molecules with the desired property. Recently, deep learning has gained considerable attention as a promising approach to solve the problem. In this paper, we propose genetic expert-guided learning (GEGL), a simple yet novel framework for training a deep neural network (DNN) to generate highly-rewarding molecules. Our main idea is to design a "genetic expert improvement" procedure, which generates high-quality targets for imitation learning of the DNN. Extensive experiments show that GEGL significantly improves over state-of-the-art methods. For example, GEGL manages to solve the penalized octanol-water partition coefficient optimization with a score of 31.40, while the best-known score in the literature is 27.22. Besides, for the GuacaMol benchmark with 20 tasks, our method achieves the highest score for 19 tasks, in comparison with state-of-the-art methods, and newly obtains the perfect score for three tasks. Our training code is available at https://github.com/sungsoo-ahn/genetic-expert-guided-learning.

## 1   Introduction

Discovering new molecules with the desired property is fundamental in chemistry, with critical applications such as drug discovery [1] and material design [2]. The task is challenging since the molecular space is vast; e.g., the number of synthesizable drug-like compounds is estimated to be around $10^{60}$ [3]. To tackle this problem, *de novo molecular design* [4, 5] aims to generate a new molecule from scratch with the desired property, rather than naïvely enumerate over the molecular space.

Over the past few years, molecule-generating deep neural networks (DNNs) have demonstrated successful results for solving the de novo molecular design problem [6, 7, 8, 9, 10, 11, 12, 13, 14, 15, 16, 17, 18, 19, 20, 21, 22]. For example, Gómez-Bombarelli et al. [8] perform Bayesian optimization for maximizing the desired property, on the embedding space of molecule-generating variational auto-encoders. On the other hand, Guimaraes et al. [6] train a molecule-generating policy using reinforcement learning with the desired property formulated as a reward.

Intriguingly, several works [23, 24, 25, 26] have recently evidenced that the traditional frameworks based on genetic algorithm (GA) can compete with or even outperform the recently proposed deep learning methods. They reveal that GA is effective, thanks to the powerful domain-specific genetic operators for exploring the chemical space. For example, Jensen [24] achieves outstanding performance by generating new molecules as a combination of subgraphs extracted from existing ones. Such observations also emphasize how domain knowledge can play a significant role in de novo molecular design. On the contrary, the current DNN-based methods do not exploit such domain knowledge explicitly; instead, they implicitly generalize the knowledge of high-rewarding molecules by training a DNN on them. Notably, the expressive power of DNN allows itself to parameterize a distribution over the whole molecular space flexibly.

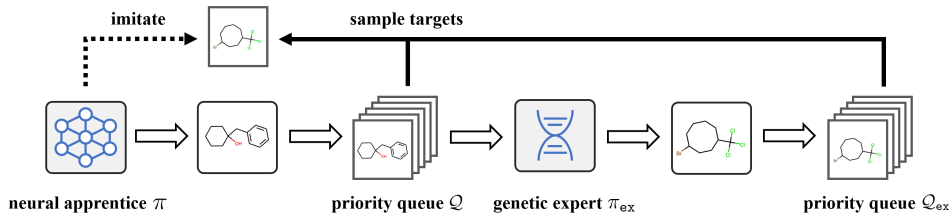

Figure 1: Illustration of the proposed genetic expert-guided learning (GEGL) framework.

**Contribution.** In this work, we propose genetic expert-guided learning (GEGL), which is a novel framework for training a molecule-generating DNN guided with genetic exploration. Our main idea is to formulate an *expert policy* by applying the domain-specific genetic operators, i.e., mutation and crossover,[1] to the DNN-generated molecules. Then the DNN becomes an *apprentice policy* that learns to imitate the highly-rewarding molecules discovered by the expert policy. Since the expert policy improves over the apprentice policy by design, the former policy consistently guides the latter policy to generate highly-rewarding molecules. We provide an overall illustration of our framework in Figure 1.

We note that our GEGL framework can be seen as a reinforcement learning algorithm with a novel mechanism for additional explorations. To be specific, the generation of a molecule can be regarded as an action and the desired property of the generated molecule as a reward. Similar to most reinforcement learning algorithms, reducing the sample complexity is crucial in our GEGL framework. To this end, we design our framework with *max-reward priority queues* [13, 27, 28, 29, 30]. By storing the highly-rewarding molecules, the priority queues prevent the policies from "forgetting" the valuable knowledge.

We extensively evaluate our method on four experiments: (a) optimization of penalized octanol-water partition coefficient, (b) optimization of penalized octanol-water partition coefficient under similarity constraints, (c) the GuacaMol benchmark [31] consisting of 20 de novo molecular design tasks, and (d) the GuacaMol benchmark evaluated under post-hoc filtering procedure [32]. Remarkably, our GEGL framework outperforms all prior methods for de novo molecular design by a large margin. In particular, GEGL achieves the penalized octanol-water partition coefficient score of 31.40, while the best baseline [22] and the second-best baseline [17] achieves the score of 27.22 and 26.1, respectively. For the GuacaMol benchmark, our algorithm achieves the highest score for 19 out of 20 tasks and newly achieves the perfect score for three tasks.

## 2 Related works

Automating the discovery of new molecules is likely to have significant impacts on essential applications such as drug discovery [1] and material design [2]. To achieve this, researchers have traditionally relied on *virtual screening* [33], which typically works in two steps: (a) enumerating all the possible combinations of predefined building-block molecules and (b) reporting the molecules with the desired property. However, the molecular space is large, and it is computationally prohibitive to enumerate and score the desired property for all the possible molecules.

*De novo molecular design* [4] methods attempt to circumvent this issue by generating a molecule from scratch. Instead of enumerating a large set of molecules, these methods search over the molecular space to maximize the desired property. In the following, we discuss the existing methods for de novo molecular design categorized by their optimization schemes: deep reinforcement learning, deep embedding optimization, and genetic algorithm.

**Deep reinforcement learning (DRL).** Deep reinforcement learning is arguably the most straightforward approach for solving the de novo molecular design problem with deep neural networks (DNNs) [6, 9, 13, 16, 18, 19, 22]. Such methods formulate the generation of the molecules as a Markov decision process. In such formulations, the desired properties of molecules correspond to high rewards, and the policy learns to generate highly-rewarding molecules. We also note several works using deep reinforcement learning to solve combinatorial problems similar to the task of de

novo molecular design, e.g., program synthesis [28], combinatorial mathematics [34], biological sequence design [35], and protein structure design [36].

**Deep embedding optimization (DEO).** Also, there exist approaches to optimize molecular "embeddings" extracted from DNNs [7, 8, 10, 11, 12, 15, 17, 20]. Such methods are based on training a neural network to learn a mapping between a continuous embedding space and a discrete molecular space. Then they apply optimization over the continuous embedding to maximize the desired property of the corresponding molecule. Methods such as gradient descent [10, 15], Bayesian optimization [7, 8, 11, 12], constrained Bayesian optimization [20], and particle swarm optimization [17] have been applied for continuous optimization of the embedding space.

**Genetic algorithm (GA).** Inspired from the concept of natural selection, genetic algorithms (GAs) [23, 24, 26, 37, 38, 39, 40, 41] search over the molecular space with genetic operators, i.e., mutation and crossover. To this end, the mutation randomly modifies an existing molecule, and the crossover randomly combines a pair of molecules. Such approaches are appealing since they are simple and capable of incorporating the domain knowledge provided by human experts. While the deep learning methods are gaining attention for de novo molecular design, several works [23, 24, 26] have recently demonstrated that GA can compete with or even outperform the DNN-based methods.

## 3 Genetic expert-guided learning (GEGL)

### 3.1 Overview of GEGL

In this section, we introduce genetic expert-guided learning (GEGL), a novel yet simple framework for de novo molecular design. To discover highly-rewarding molecules, GEGL aims at training a molecule-generating deep neural network (DNN). Especially, we design the framework using an additional genetic *expert policy*, which generates targets for imitation learning of the neural *apprentice policy*, i.e., the DNN. Our main idea is about formulating the expert policy as a "genetic improvement operator" applied to the apprentice policy; this allows us to steer the apprentice policy towards generating highly-rewarding molecules by imitating the better expert policy.

To apply our framework, we view de novo molecular design as a combinatorial optimization of discovering a molecule $x$, which maximizes the *reward* $r(x)$, i.e., the desired property.[2] To solve this problem, we collect highly-rewarding molecules from the neural apprentice policy $\pi(x; \theta)$ and the genetic expert policy $\pi_{\text{ex}}(x; \mathcal{X})$ throughout the learning process. Here, $\theta$ indicates the DNN's parameter representing the apprentice policy, and $\mathcal{X}$ indicates a set of "seed" molecules to apply the genetic operators for the expert policy. Finally, we introduce fixed-size *max-reward priority queues* $\mathcal{Q}$ and $\mathcal{Q}_{\text{ex}}$, which are buffers that only keep a fixed number of molecules with the highest rewards.

Our GEGL framework repeats the following three-step procedure:

**Step A.** The apprentice policy $\pi(x; \theta)$ generates a set of molecules. Then the max-reward priority queue $\mathcal{Q}$ with the size of $K$ stores the generated molecules.

**Step B.** The expert policy $\pi_{\text{ex}}(x; \mathcal{Q})$ generates molecules using the updated priority queue $\mathcal{Q}$ as the seed molecules. Next, the priority queue $\mathcal{Q}_{\text{ex}}$ with a size of $K$ stores the generated molecules.

**Step C.** The apprentice policy optimizes its parameter $\theta$ by learning to imitate the molecules sampled from the union of the priority queues $\mathcal{Q} \cup \mathcal{Q}_{\text{ex}}$. In other words, the parameter $\theta$ is updated to maximize $\sum_{x \in \mathcal{Q} \cup \mathcal{Q}_{\text{ex}}} \log \pi(x; \theta)$.

One may interpret GEGL as a deep reinforcement learning algorithm. From such a perspective, the corresponding Markov decision process has a fixed episode-length of one, and its action corresponds to the generation of a molecule. Furthermore, GEGL highly resembles the prior works on expert iteration [42, 43] and AlphaGo Zero [44], where the Monte Carlo tree search is used as an improvement operator that guides training through enhanced exploration. We provide an illustration and a detailed description of our algorithm in Figure 1 and Algorithm 1, respectively.

**Algorithm 1** Genetic expert-guided learning (GEGL)

1: Set $\mathcal{Q} \leftarrow \emptyset, \mathcal{Q}_{\text{ex}} \leftarrow \emptyset.$             ▷ *Initialize the max-reward priority queues $\mathcal{Q}$ and $\mathcal{Q}_{\text{ex}}$.*
2: **for** $t = 1, \ldots, T$ **do**
3:      **for** $m = 1, \ldots, M$ **do**            ▷ *Step A: add $M$ samples generated by $\pi$ into $\mathcal{Q}$.*
4:          Update $\mathcal{Q} \leftarrow \mathcal{Q} \cup \{\boldsymbol{x}\}$, where $\boldsymbol{x} \sim \pi(\boldsymbol{x}; \theta)$.
5:          If $|\mathcal{Q}| > K$, update $\mathcal{Q} \leftarrow \mathcal{Q} \setminus \{\boldsymbol{x}_{\min}\}$, where $\boldsymbol{x}_{\min} = \arg\min_{\boldsymbol{x} \in \mathcal{Q}} r(\boldsymbol{x})$.
6:      **end for**
7:      **for** $m = 1, \ldots, M$ **do**            ▷ *Step B: add $M$ samples generated by $\pi_{\text{ex}}$ into $\mathcal{Q}_{\text{ex}}$.*
8:          Update $\mathcal{Q}_{\text{ex}} \leftarrow \mathcal{Q}_{\text{ex}} \cup \{\boldsymbol{x}\}$, where $\boldsymbol{x} \sim \pi_{\text{ex}}(\boldsymbol{x}; \mathcal{Q})$.
9:          If $|\mathcal{Q}_{\text{ex}}| > K$, update $\mathcal{Q}_{\text{ex}} \leftarrow \mathcal{Q}_{\text{ex}} \setminus \{\boldsymbol{x}_{\min}\}$, where $\boldsymbol{x}_{\min} = \arg\min_{\boldsymbol{x} \in \mathcal{Q}_{\text{ex}}} r(\boldsymbol{x})$.
10:     **end for**
11:     Maximize $\sum_{\boldsymbol{x} \in \mathcal{Q} \cup \mathcal{Q}_{\text{ex}}} \log \pi(\boldsymbol{x}; \theta)$ over $\theta$.      ▷ *Step C: train $\pi$ with imitation learning.*
12: **end for**
13: Report $\mathcal{Q} \cup \mathcal{Q}_{\text{ex}}$ as the output.          ▷ *Output the highly-rewarding molecules.*

## 3.2 Detailed components of GEGL

In the rest of this section, we provide a detailed description of each component in GEGL. We start by explaining our design choices on the expert and the apprentice policies. Then we discuss how the max-reward priority queues play an essential role in our framework.

**Genetic expert policy.** Our genetic expert policy $\pi_{\text{ex}}(x; \mathcal{X})$ is a distribution induced by applying the genetic operators, i.e., mutation and crossover, to a set of molecules $\mathcal{X}$. We use the genetic operators highly optimized (with domain knowledge) for searching over the molecular space; hence the expert policy efficiently improves over the apprentice policy in terms of exploration.

An adequate choice of genetic operators is crucial for the expert policy. To this end, we choose the graph-based mutation and crossover proposed by Jensen [24], as they recently demonstrated outstanding performance for de novo molecular design. At a high-level, the genetic expert policy $\pi_{\text{ex}}(x; \mathcal{X})$ generates a molecule in two steps. First, the expert policy generates a *child* molecule by applying the crossover to a pair of *parent* molecules randomly drawn from $\mathcal{X}$. To be specific, two subgraphs extracted from the parents attach to form a child molecule. We use two types of operations for crossover: `non_ring_crossover` and `ring_crossover`. Next, with a small probability, the expert policy mutates the child by atom-wise or bond-wise modification, e.g., adding an atom. We use seven types of operations for mutation: `atom_deletion`, `atom_addition`, `atom_insertion`, `atom_type_change`, `bond_order_change`, `ring_bond_deletion`, and `ring_bond_addition`. We provide an illustration and a detailed description of the genetic operators in Figure 2 and Appendix A, respectively.

We also note that other choices of the improvement operators are possible for the expert policy. Especially, Gao and Coley [32] demonstrated how naïvely applying the genetic operators may lead to proposing molecules that are unstable or unsynthesizable in real world. To consider this aspect, one may use exploration operators that considers the chemical validity of the molecules. For example, Polishchuk [26] proposed a genetic algorithm based on "chemically reasonable" mutations. In addition, Bradshaw et al. [15] proposed to generate molecules based on reaction models, which could also be used as an improvement operator in our framework.

**Neural apprentice policy.** We parameterize our neural apprentice policy using a long-short term memory (LSTM) network [45]. Moreover, the molecules are represented by a sequence of characters, i.e., the simplified molecular-input line-entry system (SMILES) [46] format. Under such a design, the probability $\pi(\boldsymbol{x}; \theta)$ of a molecule $\boldsymbol{x}$ being generated from the apprentice policy is factorized into $\prod_{n=1}^{N} \pi(x_n | x_1, \ldots, x_{n-1}; \theta)$. Here, $x_1, \ldots, x_N$ are the characters corresponding to canonical SMILES representation of the given molecule.

We note that our choice of using the LSTM network to generate SMILES representations of molecules might not seem evident. Especially, molecular graph representations alternatively express the molecule, and many works have proposed new molecule-generating graph neural networks (GNNs) [8, 12, 16, 19, 47, 48]. However, no particular GNN architecture clearly dominates over others, and recent molecular generation benchmarks [31, 49] report that the LSTM network matches (or improves

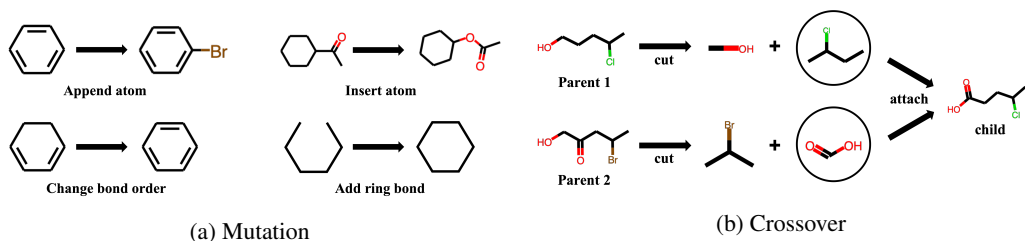

(a) Mutation                                   (b) Crossover

Figure 2: Illustration of (a) mutation and (b) crossover used in the genetic expert policy.

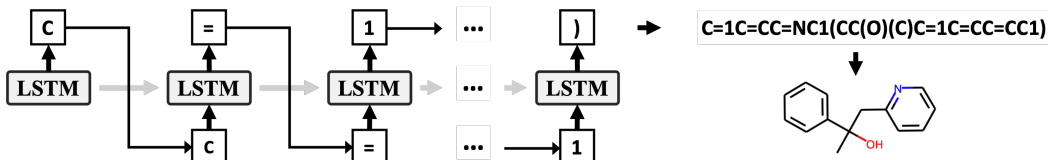

Figure 3: Illustration of the apprentice policy generating a SMILES representation of a molecule.

over) the performance of GNNs. Therefore, while searching for the best molecule-generating DNN architecture is a significant research direction, we leave it for future work. Instead, we choose the well-established LSTM architecture for the apprentice policy.

**Max-reward priority queues.** Role of the max-reward priority queues $\mathcal{Q}$ and $\mathcal{Q}_{\text{ex}}$ in our framework is twofold. First, the priority queues provide highly-rewarding molecules for the expert and the apprentice policy. Furthermore, they prevent the policies from "forgetting" the highly-rewarding molecules observed in prior. Recent works [13, 27, 28, 29, 30] have also shown similar concepts to be successful for learning in deterministic environments.

We further elaborate on our choice of training the apprentice policy on the union of the priority queues $\mathcal{Q} \cup \mathcal{Q}_{\text{ex}}$ instead of the priority queue $\mathcal{Q}_{\text{ex}}$. This choice comes from how the expert policy does not always improve the apprentice policy in terms of reward, although it does in terms of exploration. Hence, it is beneficial for the apprentice policy to imitate the highly-rewarding molecules generated from both the apprentice and the expert policy. This promotes the apprentice to be trained on molecules with improved rewards.

## 4 Experiments

In this section, we report the experimental results of the proposed genetic expert-guided learning (GEGL) framework. To this end, we extensively compare GEGL with the existing works, for the optimization of the penalized octanol-water partition coefficient [50] and the GuacaMol benchmark [31]. We also provide additional experimental results in Appendix B; these results consist of relatively straightforward tasks such as targeted optimization of the octanol-water partition coefficient and optimization of the quantitative estimate of drug-likeness (QED) [51]. For comparison, we consider a variety of existing works on de novo molecular design based on deep reinforcement learning (DRL), deep embedding optimization (DEO), genetic algorithm (GA), and deep supervised learning (DSL).[3] We report the numbers obtained by the existing works unless stated otherwise. All of the experiments were processed using single GPU (NVIDIA RTX 2080Ti) and eight instances from a virtual CPU system (Intel Xeon E5-2630 v4). We also provide descriptions the employed baselines in Appendix C. A detailed illustration for the molecules generated by GEGL appears in Appendix D.

**Implementation details.** To implement GEGL, we use priority queue of fixed size $K = 1024$. At each step, we sample 8192 molecules from the apprentice and the expert policy to update the respective priority queues. Adam optimizer [52] with learning rate of 0.001 was used to optimize the neural network with a mini-batch of size 256. Gradients were clipped by a norm of 1.0. The apprentice policy is constructed using three-layered LSTM associated with hidden state of 1024 dimensions and dropout probability of 0.2. At each step of GEGL, the apprentice policy generates

Table 1: Experimental results on the optimization of (a) `PenalizedLogP` and (b) `PenalizedLogP` with similarity constraint of $\delta$. Types of the algorithms are indicated by deep reinforcement learning (DRL), deep embedding optimization (DEO), deep supervised learning (DSL), and genetic algorithm (GA). [†‡]We report the average and standard deviation of the objectives collected over five independent runs and 800 molecules for (a) and (b), respectively.

(a) `PenalizedLogP`

| Algorithm | Type | Objective |
|---|---|---|
| GVAE+BO [Kusner et al. 2017] | DEO | $2.87 \pm 0.06$ |
| SD-VAE [Dai et al. 2018] | DEO | $3.50 \pm 0.44$ |
| ORGAN [Guimaraes et al. 2017] | DRL | $3.52 \pm 0.08$ |
| VAE+CBO [Griffiths and Hernández-Lobato 2020] | DEO | $4.01$ |
| ChemGE [Yoshikawa et al. 2018] | GA | $4.53 \pm 0.26$ |
| CVAE+BO [Gómez-Bombarelli et al. 2018] | DEO | $4.85 \pm 0.17$ |
| JT-VAE [Jin et al. 2018] | DEO | $4.90 \pm 0.33$ |
| ChemTS [Yang et al. 2017] | DRL | $5.6 \pm 0.5$ |
| GCPN [You et al. 2018] | DRL | $7.86 \pm 0.07$ |
| MRNN [Popova et al. 2019] | DRL | $8.63$ |
| MolDQN [Zhou et al. 2019] | DRL | $11.84$ |
| GraphAF [Shi et al. 2020] | DRL | $12.23$ |
| GB-GA [Jensen 2019] | GA | $15.76 \pm 5.76$ |
| DA-GA [Nigam et al. 2020] | GA | $20.72 \pm 3.14$ |
| MSO [Winter et al. 2019] | DEO | $26.1$ |
| PGFS [Gottipati et al. 2020] | DRL | $27.22$ |
| GEGL[†] (Ours) | DRL | $\mathbf{31.40} \pm \mathbf{0.00}$ |

(b) Similarity-constrained `PenalizedLogP`

| $\delta$ | Algorithm | Type | Objective | Succ. rate |
|---|---|---|---|---|
| 0.4 | JT-VAE [Jin et al. 2018] | DEO | $0.84 \pm 1.45$ | 0.84 |
| | GCPN [You et al. 2018] | DRL | $2.49 \pm 1.30$ | 1.00 |
| | DEFactor [Assouel et al. 2018] | DEO | $3.41 \pm 1.67$ | 0.86 |
| | VJTNN [Jin et al. 2019] | DSL | $3.55 \pm 1.67$ | - |
| | HierG2G [Jin et al. 2020] | DSL | $3.98 \pm 1.09$ | - |
| | GEGL[‡] (Ours) | DRL | $\mathbf{7.87} \pm \mathbf{1.81}$ | 1.00 |
| 0.6 | JT-VAE [Jin et al. 2018] | DEO | $0.21 \pm 0.71$ | 0.47 |
| | GCPN [You et al. 2018] | DRL | $0.79 \pm 0.63$ | 1.00 |
| | DEFactor [Assouel et al. 2018] | DEO | $1.55 \pm 1.19$ | 0.73 |
| | VJTNN [Jin et al. 2019] | DSL | $2.33 \pm 1.17$ | - |
| | HierG2G [Jin et al. 2020] | DSL | $2.49 \pm 1.46$ | - |
| | GEGL[‡] (Ours) | DRL | $\mathbf{4.43} \pm \mathbf{1.53}$ | 1.00 |

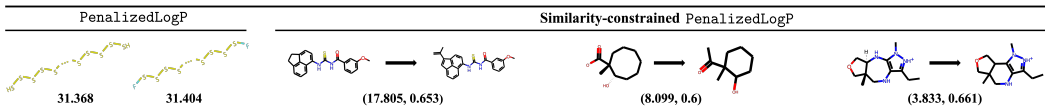

| `PenalizedLogP` | | Similarity-constrained `PenalizedLogP` | | |
|---|---|---|---|---|
| 31.368 | 31.404 | (17.805, 0.653) | (8.099, 0.6) | (3.833, 0.661) |

Figure 4: Illustration of the highly-scoring molecule and (reference molecule, improved molecule) for the unconstrained and similarity-constrained `PenalizedLogP` optimization, respectively. Below each molecule, we denote the associated objective and (objective, similarity) for the unconstrained and similarity-constrained `PenalizedLogP` optimization, respectively.

8192 molecules. Among the generated samples, *invalid* molecules, e.g., molecules violating the valency rules, are filtered out from output of the apprentice policy. Next, the expert policy generates a molecule by selecting 8192 pair of molecules from the priority queue to apply crossover. For each valid molecules generated from crossover operation, mutation is applied with probability of 0.01. Similar to the apprentice policy, invalid molecules are filtered out from output of the expert policy.

### 4.1 Optimization of penalized octanol-water partition coefficient

Comparing to the literature standard, we aim at maximizing the *penalized octanol-water partition coefficient* (`PenalizedLogP`) score defined as follows:

$$\texttt{PenalizedLogP}(\boldsymbol{x}) = \texttt{LogP}(\boldsymbol{x}) - \texttt{SyntheticAccessibility}(\boldsymbol{x}) - \texttt{RingPenalty}(\boldsymbol{x}),$$

where `LogP`, `SyntheticAccessibility`, and `RingPenalty` correspond to the (unpenalized) octanol-water partition coefficient [55], the synthetic accessibility [56] penalty, and the penalty for atom rings of size larger than 6. We also note that Kusner et al. [7] first performed this task motivated by how the octanol-water partition coefficient plays an important role in characterizing the drug-likeness of a molecule. Following prior works [12, 9], we pretrain the apprentice policy on the ZINC dataset [57].

**Unconstrained optimization.** First, we attempt to find a molecule maximizing the `PenalizedLogP` score as the objective without specific constraints. To this end, we run GEGL for 200 steps while limiting the generation to molecules with at most 81 SMILES characters (as done by Jensen [24], Nigam et al. [25], and Yang et al. [53]).

In Table 1a, we observe that our algorithm indeed outperforms the existing baselines by a large margin. In particular, GEGL achieves a score of 31.40, which relatively outperforms the second-best

(PGFS) and the third-best (MSO) baselines by $15\%$ and $20\%$, respectively. This result highlights the strong performance of GEGL.

**Constrained optimization.** In this experiment, we follow the experimental setup proposed by Jin et al. [12]. Namely, for each molecule $x_{\texttt{ref}}$ from 800 low-`PenalizedLogP`-scoring molecules from the ZINC data set [57], we search for a new molecule $x$ with the maximum `PenalizedLogP` score while being constrained to be similar to the reference molecule $x_{\texttt{ref}}$. To be specific, we express this optimization as follows:

$$\max_{x} \texttt{PenalizedLogP}(x) - \texttt{PenalizedLogP}(x_{\texttt{ref}}), \quad \text{s.t.} \quad \texttt{Similarity}(x, x_{\texttt{ref}}) \geq \delta,$$

where $\texttt{Similarity}(\cdot, \cdot)$ is the Tanimoto similarity score [58]. In the experiments, we run GEGL for 50 steps for each reference molecule $x_{\texttt{ref}}$. We initialize the apprentice's priority queue $\mathcal{Q}$ with the reference molecule $x_{\texttt{ref}}$, i.e., $\mathcal{Q} \leftarrow \{x_{\texttt{ref}}\}$. For this experiment, we constrain the maximum length of SMILES to be 100. We report the above objective averaged over the 800 molecules. We also evaluate the "success ratio" of the algorithms, i.e., the ratio of molecules with a positive objective.

In Table 1b, we once again observe GEGL to achieve superior performance to the existing algorithms. We also note that our algorithm always succeeds in improving the `PenalizedLogP` score of the reference molecule, i.e., the success ratio is 1.00.

**Generated molecules.** We now report the molecules generated for unconstrained and constrained optimization of `PenalizedLogP` score in Figure 4. Notably, for unconstrained optimization, we observe that our model produces "unrealistic" molecules that contain a long chain of sulfurs. This symptom arise from our method exploiting the ambiguity of the `PenalizedLogP` score, i.e., the score spuriously assigning high values to the unrealistic molecules. Indeed, similar observations (on algorithms generating a long chain of sulfurs) have been made regardless of optimization methods for de novo molecular design. For example, Shi et al. [19], Winter et al. [17], and Nigam et al. [25] reported molecules with similar structure when using deep reinforcement learning, deep embedding optimization, and genetic algorithm, respectively. Such an observation emphasizes why one should carefully design the procedure of de novo molecular design.

## 4.2 GuacaMol benchmark

Next, we provide the empirical results for the GuacaMol benchmark [31], designed specifically to measure the performance of de novo molecular design algorithms. It consists of 20 chemically meaningful molecular design tasks, that have been carefully designed and studied by domain-experts in the past literature [41, 47, 59, 60, 61]. Notably, the GuacaMol benchmark scores a set of molecules rather than a single one, to evaluate the algorithms' ability to produce diverse molecules. To this end, given a set of molecules $\mathcal{X} = \{x_s\}_{s=1}^{|\mathcal{X}|}$ and a set of positive integers $\mathcal{S}$, tasks in the GuacaMol benchmark evaluate their score as follows:

$$\texttt{GuacaMolScore}(\mathcal{X}) \coloneqq \sum_{S \in \mathcal{S}} \sum_{s=1}^{S} \frac{r(x_{\Pi(s)})}{S|\mathcal{S}|}, \quad \text{where } r(x_{\Pi(s)}) \geq r(x_{\Pi(s+1)}) \text{ for } s = 1, \ldots, |\mathcal{X}|-1,$$

where $r$ is the task-specific molecule-wise score and $\Pi$ denotes a permutation that sorts the set of molecules $\mathcal{X}$ in descending order of their metrics. We further provide details on the benchmark in Appendix E.

For the experiments, we initialize the apprentice policy using the weights provided by Brown et al. [31],[4] that was pretrained on the ChEMBL [62] dataset. For each tasks, we run GEGL for 200 steps. We constrain the maximum length of SMILES to be 100.

**Benchmark results.** In Table 2, we observe that GEGL outperforms the existing baselines by a large margin. Namely, GEGL achieves the highest score for 19 out of 20 tasks. Furthermore, our algorithm perfectly solves thirteen tasks,[5] where three of them were not known to be perfectly solved. Such a result demonstrates how our algorithm effectively produces a high-rewarding and diverse set of molecules.

**Evaluation with post-hoc filtering.** As observed in Section 4.1 and prior works, de novo molecular design algorithms may lead to problematic results. For example, the generated molecules may be

Table 2: Experimental results for task-ids of $1, 2, \ldots, 20$ from (a) the GuacaMol benchmark and (b) the GuacaMol benchmark evaluated with post-hoc filtering process. See Appendix E for the description of each task per id. *We re-evaluate the official implementation for baselines in (b).

|  | (a) GuacaMol | | | | | | | | (b) GuacaMol with filtering | | | | |
|---|---|---|---|---|---|---|---|---|---|---|---|---|---|
| id | ChEMBL [62] | MCTS [24] | ChemGE [23] | HC-MLE [13] | GB-GA [24] | MSO [17] | CReM [26] | GEGL (Ours) | ChEMBL* [62] | ChemGE* [23] | HC-MLE* [13] | GB-GA* [24] | GEGL (Ours) |
| 1 | 0.505 | 0.355 | 0.732 | **1.000** | **1.000** | **1.000** | **1.000** | **1.000** | 0.505 | 0.646 | **1.000** | **1.000** | **1.000** |
| 2 | 0.418 | 0.311 | 0.515 | **1.000** | **1.000** | **1.000** | **1.000** | **1.000** | 0.260 | 0.504 | 0.537 | **0.837** | 0.552 |
| 3 | 0.456 | 0.311 | 0.598 | **1.000** | **1.000** | **1.000** | **1.000** | **1.000** | 0.456 | 0.552 | **1.000** | **1.000** | **1.000** |
| 4 | 0.595 | 0.380 | 0.834 | **1.000** | **1.000** | **1.000** | **1.000** | **1.000** | 0.595 | 0.769 | **1.000** | 0.995 | **1.000** |
| 5 | 0.719 | 0.749 | 0.907 | **1.000** | **1.000** | **1.000** | **1.000** | **1.000** | 0.711 | 0.959 | **1.000** | 0.996 | **1.000** |
| 6 | 0.629 | 0.402 | 0.790 | **1.000** | **1.000** | **1.000** | **1.000** | **1.000** | 0.632 | 0.631 | **1.000** | 0.996 | **1.000** |
| 7 | 0.684 | 0.410 | 0.829 | 0.993 | 0.971 | 0.997 | 0.966 | **1.000** | 0.684 | 0.786 | 0.997 | 0.960 | **1.000** |
| 8 | 0.747 | 0.632 | 0.889 | 0.879 | 0.982 | **1.000** | 0.940 | **1.000** | 0.747 | 0.883 | 0.992 | 0.823 | **1.000** |
| 9 | 0.334 | 0.225 | 0.334 | 0.438 | 0.406 | 0.437 | 0.371 | **0.455** | 0.334 | 0.361 | 0.453 | 0.402 | **0.455** |
| 10 | 0.351 | 0.170 | 0.380 | 0.422 | 0.432 | 0.395 | 0.434 | **0.437** | 0.351 | 0.377 | 0.433 | 0.420 | **0.437** |
| 11 | 0.839 | 0.784 | 0.886 | 0.907 | 0.953 | 0.966 | 0.995 | **1.000** | 0.839 | 0.895 | 0.916 | 0.914 | **1.000** |
| 12 | 0.817 | 0.695 | 0.931 | 0.959 | 0.998 | **1.000** | **1.000** | **1.000** | 0.815 | 0.920 | 0.999 | 0.905 | **1.000** |
| 13 | 0.792 | 0.616 | 0.881 | 0.855 | 0.920 | 0.931 | **0.969** | 0.958 | 0.786 | 0.714 | 0.882 | 0.530 | **0.933** |
| 14 | 0.575 | 0.385 | 0.661 | 0.808 | 0.792 | 0.834 | 0.815 | **0.882** | 0.572 | 0.572 | 0.835 | 0.780 | **0.833** |
| 15 | 0.696 | 0.533 | 0.722 | 0.894 | 0.894 | 0.900 | 0.902 | **0.924** | 0.679 | 0.709 | 0.902 | 0.889 | **0.905** |
| 16 | 0.509 | 0.458 | 0.689 | 0.545 | 0.891 | 0.868 | 0.763 | **0.922** | 0.501 | 0.587 | 0.601 | 0.634 | **0.749** |
| 17 | 0.547 | 0.488 | 0.413 | 0.669 | 0.754 | 0.764 | 0.770 | **0.834** | 0.547 | 0.647 | 0.715 | 0.698 | **0.763** |
| 18 | 0.259 | 0.040 | 0.552 | 0.978 | 0.990 | 0.994 | 0.994 | **1.000** | 0.127 | 0.827 | 0.992 | 0.789 | **1.000** |
| 19 | 0.933 | 0.590 | 0.970 | 0.996 | **1.000** | **1.000** | **1.000** | **1.000** | 0.933 | 0.857 | **1.000** | 0.994 | **1.000** |
| 20 | 0.738 | 0.470 | 0.885 | 0.998 | **1.000** | **1.000** | **1.000** | **1.000** | 0.690 | 0.964 | **1.000** | **1.000** | **1.000** |

| Albuterol similarity | | Ranolazine MPO | | Sitagliptin MPO | | Zaleplon MPO | |
|---|---|---|---|---|---|---|---|
| passed | rejected | passed | rejected | passed | rejected | passed | rejected |

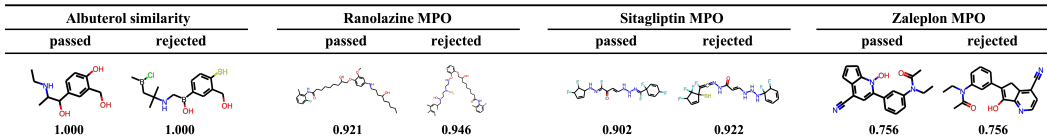

| 1.000 | 1.000 | 0.921 | 0.946 | 0.902 | 0.922 | 0.756 | 0.756 |

Figure 5: Illustration of the molecules generated for the GuacaMol benchmark, that have passed and rejected from the filtering procedure. Below each molecule, we also denote the associated objectives.

chemically reactive, hard to synthesize, or perceived to be "unrealistic" to domain experts. To consider this aspect, we evaluate our algorithm under the post-hoc filtering approach [32]. Specifically, we use the expert-designed filter to reject the molecules with undesirable feature. In other words, we train the models as in Table 2, but only report the performance of generated molecules that pass the filter. Since the filtering procedure is post-hoc, the de novo molecular design algorithms will not be able to aggressively exploit possible ambiguities of the filtering process.

As shown in Table 2b, GEGL still outperforms the baselines even when the undesirable molecules are filtered out. This validates the ability of our algorithm to generate chemically meaningful results. Hence, we conclude that our GEGL can be used flexibly with various choice of de novo molecular design process.

**Generated molecules.** In Figure 5, we illustrate the high-scoring molecules generated by GEGL for the Albuterol similarity, Ranolazine MPO, Sitagliptin MPO, and Zaleplon MPO tasks from the GuacaMol benchmark (corresponding to task-ids of 4, 13, 15, 16 in Table 2). Note that the filters used in Table 2b provide reasons for rejection of the molecules. For example, the high-scoring molecule from the Zaleplon MPO task was rejected due to containing a SMILES of C=CO, i.e., *enol*. This is undesirable, as many enols have been shown to be reactive [63]. See Appendix F for further explanation of the molecules being passed and rejected from the post-hoc filtering process.

### 4.3 Ablation studies

Finally, we perform ablation studies on our algorithm to investigate the behavior of each component. To this end, we conduct experiments on the Sitagliptin MPO and Zaleplon MPO tasks from the GuacaMol benchmark. Note that Sitagliptin MPO and Zaleplon MPO tasks corresponds to task-ids of 15 and 16 in Table 2, respectively.

**Contribution from the DNN and the genetic operator.** We start by inspecting the contribution of the DNN and the genetic operator in our algorithm. To this end, we compare GEGL with (a)

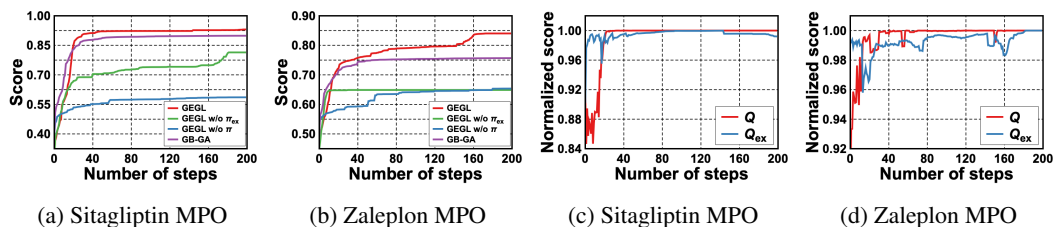

| (a) Sitagliptin MPO | (b) Zaleplon MPO | (c) Sitagliptin MPO | (d) Zaleplon MPO |

Figure 6: Illustration of ablation studies for (a, b) investigating contribution from DNN and genetic operator, and (c, d) separate evaluation of max-reward priority queues.

GEGL without the expert policy $\pi_{ex}$, (b) GEGL without the apprentice policy $\pi$, and (c) the genetic algorithm using our improvement operators, i.e., GB-GA. To be specific, (a) trains the apprentice policy to imitate the highly-rewarding molecules generated by itself. On the other hand, (b) freezes the max-reward priority queue $\mathcal{Q}$ by the highly-rewarding samples from the ChEMBL dataset [62], then only updates $\mathcal{Q}_{ex}$ based on the expert policy. Finally, (c) is the same algorithm reported in Table 2, but using the hyper-parameter of our genetic expert. In Figure 6a and 6b, we observe that all the ablation algorithms perform worse than GEGL. This result confirms that the neural apprentice policy and the genetic expert policy bring mutual benefits to our framework.

**Separate evaluation of the max-reward priority queues.** Next, we describe behavior of the apprentice policy and the expert policy during training. To this end, we compare the GuacaMol scores for the priority queues $\mathcal{Q}$ and $\mathcal{Q}_{ex}$ that are normalized by the original GEGL score. For example, we consider $\frac{\texttt{GuacaMolScore}(\mathcal{Q})}{\texttt{GuacaMolScore}(\mathcal{Q} \cup \mathcal{Q}_{ex})}$ for evaluating $\mathcal{Q}$ where $\texttt{GuacaMolScore}(\cdot)$ is the GaucaMol score evaluated on a set of molecules.

In Figure 6c and 6d, we observe that the samples collected from the genetic expert policy, i.e., $\mathcal{Q}_{ex}$, indeed improves over that of the apprentice policy, i.e., $\mathcal{Q}$ during early stages of the training. However, as the training goes on, the apprentice policy learns to generate molecules with quality higher than that of the expert policy. Since Table 6a and 6b shows that apprentice policy cannot reach the same performance without the expert policy, one may conclude that the apprentice policy effectively learns to comprise the benefits of genetic operators through learning.

## 5  Conclusion

We propose a new framework based on deep neural networks (DNNs) to solve the de novo molecular design problem. Our main idea is to enhance the training of DNN with domain knowledge, using the powerful expert-designed genetic operators to guide the training of the DNN. Through extensive experiments, our algorithm demonstrates state-of-the-art performance across a variety of tasks. We believe extending our framework to combinatorial search problems where strong genetic operators exist, e.g., biological sequence design [64], program synthesis [65], and vehicle routing problems [66], would be both promising and interesting.

## Broader Impact

**De novo molecular design.** Our framework is likely to advance the field of de novo molecular design. In this field, successful algorithms significantly impact real-world, since the discovery of a new molecule has been the key challenge of many applications. Domain of such applications includes, but are not limited to, drug molecules [67], organic light emitting diodes [68], organic solar cells [69], energetic materials [70], and electrochromic devices [71]. Improvements in these applications are beneficial to human kind in general, as they often improve the quality of human life and may broaden our knowledge of chemistry.

**Combinatorial optimization with deep reinforcement learning.** In a broader sense, our framework offers a new paradigm to search over a intractably large space of combinatorial objects with DNN. In particular, our algorithm is expected to perform well for domains where genetic algorithms are powerful; this includes domains of biological sequence design [64], program synthesis [65], and vehicle routing problems [66]. Hence, at a high-level, our work also shares the domain of applications impacted from such works.

## Acknowledgments and Disclosure of Funding

We thank Yeonghun Kang, Seonyul Kim, Jaeho Lee, Sejun Park, and Sihyun You for providing helpful feedbacks and suggestions in preparing the early version of the manuscript. This work was supported by Samsung Advanced Institute of Technology (SAIT). This work was partly supported by Institute of Information & Communications Technology Planning & Evaluation (IITP) grant funded by the Korea government (MSIT) (No.2019-0-00075, Artificial Intelligence Graduate School Program (KAIST)).

## Footnotes

[1]See Figure 2 for illustrations of mutation and crossover.

[2]In this paper, we consider the properties of molecules which can be measured quantitatively.

[3]DSL-based methods train DNNs to imitate highly-rewarding molecules provided as supervisions.

[4]https://github.com/BenevolentAI/guacamol_baselines

[5]Every score is normalized to be in the range of $[0, 1]$.

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
