[Supplementary Material]

# A  Details of the genetic operators

In this paper, we use crossover and mutation proposed by Jensen [24] for exploring the chemical space.

**Crossover.** The crossover randomly applies either `non_ring_crossover` or `ring_crossover`, with equal probability. This generates two (possibly invalid) child molecules. If both the child molecules are invalid, e.g., violating the valency rules, the crossover is re-applied with limited number of trials. If valid molecules exist, the we choose one of them randomly. In the following, we provide further details on the `non_ring_crossover` and `ring_crossover`.

    **a.** The `non_ring_crossover` cuts an arbitrary edge, which does not belong to ring, of two parent molecules, and then attach the subgraphs from different parent molecules.

    **b.** The `ring_crossover` cuts two edges in an arbitrary ring, and attach the subgraphs from different parent molecules.

**Mutation.** The mutation randomly applies one of the seven different ways for modifying a molecule: `atom_deletion`, `atom_addition`, `atom_insertion`, `atom_type_change`, `ring_bond_deletion`, `ring_bond_addition`, and `bond_order_change`. After mutation, if the modified molecule is not valid, we discard it and re-apply mutation. Details of seven different ways of modifying a molecule are as follows.

    **a.** The `atom_deletion` removes a single atom and rearrange neighbor molecules with minimal deformation from the original molecule.

    **b.** The `atom_addition` connects a new atom to a single atom.

    **c.** The `atom_insertion` puts an atom between two atoms.

    **d.** The `atom_type_change` newly changes a type of an atom.

    **e.** The `bond_order_change` alters the type of a bond.

    **f.** The `ring_bond_deletion` cuts a bond from ring.

    **g.** The `ring_bond_addition` creates a "shortcut" between two connected atoms.

# B Additional experiments

Table 3: Experiment results for quantitative estimate of drug-likeness (QED) task.

| Algorithm | Type | Objective |
|---|---|---|
| MCTS [Jensen 2019] | MCTS | 0.851 |
| ORGAN [Guimaraes et al. 2017] | DRL | 0.896 |
| JT-VAE [Jin et al. 2018] | DEO | 0.925 |
| ChemGE [Yoshikawa et al. 2018] | GA | **0.948** |
| GCPN [You et al. 2018] | DRL | **0.948** |
| MRNN [Popova et al. 2019] | DRL | **0.948** |
| MolDQN [Zhou et al. 2019] | DRL | **0.948** |
| GraphAF [Shi et al. 2020] | DRL | **0.948** |
| GB-GA [Jensen 2019] | GA | **0.948** |
| HC-MLE [Jensen 2019] | DRL | **0.948** |
| MSO [Winter et al. 2019] | DEO | **0.948** |
| GEGL[†] (Ours) | DRL | **0.948** |

Table 4: Experimental results on relatively straight-forward tasks from the Guacamol benchmark.

| Task | ChEMBL [62] | MCTS [24] | ChemGE [23] | HC-MLE [13] | GB-GA [24] | GEGL (Ours) |
|---|---|---|---|---|---|---|
| logP (target: -1.0) | **1.000** | **1.000** | **1.000** | **1.000** | **1.000** | **1.000** |
| logP (target: 8.0) | **1.000** | **1.000** | **1.000** | **1.000** | **1.000** | **1.000** |
| TPSA (target: 150.0) | **1.000** | **1.000** | **1.000** | **1.000** | **1.000** | **1.000** |
| CNS MPO | **1.000** | **1.000** | **1.000** | **1.000** | **1.000** | **1.000** |
| $C_7H_8N_2O_2$ | 0.972 | 0.851 | 0.992 | **1.000** | 0.993 | **1.000** |
| Pioglitazone MPO | 0.982 | 0.941 | **1.000** | 0.993 | 0.998 | 0.999 |

In this section, we provide additional experimental results for the tasks in the GuacaMol benchmark Brown et al. [31] that were determined to be relatively more straight-forward than other tasks. To this end, we first report our result for unconstrained optimization of the quantitative estimate of drug-likeness (QED) [51] in Table 3. Next, we evaluate GEGL on other tasks from the GaucaMol benchmark in Table 4. In the Table 3 and 4, we observe GEGL to achieve the highest scores for six out of seven tasks. See Appendix E for details on the tasks considered in this section.

# C  Baselines for de novo molecular design

In this section, we briefly describe the algorithms we used as baselines for evaluating our algorithm.

1. GVAE+BO [7] trains a *grammar* variational autoencoder and applies Bayesian optimization on its embedding space.

2. SD-VAE [11] propose a *syntax directed* variational autoencoder and applies Bayesian optimization on its embedding space.

3. ORGAN [6] trains a generative adversarial network along with reinforcement learning for maximizing the object.

4. VAE+CBO [20] trains a molecule-generating variational autoencoder and applies constrained Bayesian optimization on its embedding space.

5. CVAE+BO [8] trains a *constrained* variational autoencoder and applies Bayesian optimization on its embedding space.

6. JT-VAE [12] trains a *junction tree* variational autoencoder and applies Bayesian optimization on its embedding space.

7. ChemTS [53] proposes a SMILES-based genetic algorithm.

8. GCPN [9] trains a policy parameterized with graph convolutional network with reinforcement learning to generate highly-rewarding molecules. The reward is defined as a linear combination of the desired property of the molecule and a discriminator term for generating "realistic" molecules.

9. Molecular recurrent neural network (MRNN) [16] trains a recurrent neural network using reinforcement learning.

10. MolDQN [18] trains a molecular fingerprint-based policy with reinforcement learning.

11. GraphAF [19] trains a auto-regressive flow model with reinforcement learning.

12. GB-GA [24] proposes a graph-based genetic algorithm.

12. MCTS [24] proposes a Monte Carlo tree search algorithm.

13. MSO [17] applies particle swarm optimization on the embedding space of a variational autoencoder.

14. DEFactor [14] trains a variational autoencoder where computational efficiency was enhanced with differentiable edge variables.

15. VJTNN [54] trains a graph-to-graph model with supervised learning on the highly-rewarding set of molecules.

16. HC-MLE [13] proposes a "hill-climbing" variant of reinforcement learning to train a recurrent network.

17. CReM [26] proposes a genetic algorithm based on "chemically reasonable" genetic operator to generate a set of chemically reasonable molecules.

18. HierG2G [21] trains a graph-to-graph, hierarchical generative model with supervised learning on the highly-rewarding set of molecules.

19. DA-GA [25] proposes a genetic algorithm based on its fitness function augmented with a discriminator which assigns higher scores to "novel" elements.

20. PGFS [22] proposes a reinforcement learning algorithm that generates molecules from deciding the synthesis route given the set of available reactants.

# D  Additional Illustration of the Generated Molecules

Figure 7: Additional illustration of highly-scoring molecules for the unconstrained `PenalizedLogP` optimization. Below each molecule, we denote the associated objective.

Figure 8: Additional illustration of highly-scoring molecules for the similarity-constrained `PenalizedLogP` optimization. Below each molecule, we denote the associated (objective, similarity).

Figure 9: Additional illustration of the molecules generated for the GuacaMol benchmark, that have passed and rejected from the filtering procedure. Below each molecule, we denote the associated objective.

# E Details on the GuacaMol benchmark

In this section, we further provide details on the tasks considered in the GuacaMol benchmark.

- **Rediscovery.** The {celecoxib, troglitazone, thiothixene} rediscovery tasks desire the molecules to be as similar as possible to the target molecules. The algorithms achieve the maximum score when they produce a molecule identical to the target molecule. These benchmarks have been studied by Zaliani et al. [59] and Segler et al. [47].

- **Similarity.** The {aripiprazole, albuterol, mestranol} similarity tasks also aim at finding a molecule similar to the target molecule. It is different from the rediscovery task since the metric is evaluated over multiple molecules. This similarity metric has been studied by Willett et al. [60].

- **Isometry.** The tasks of $C_{11}H_{24}$, $C_9H_{10}N_2O_2PF_2Cl$, $C_7H_8N_2O_2$ attempt to find molecules with the target formula. The algorithms achieve an optimal score when they find all of the possible isometric molecules for the tasks.

- **Median molecules.** The median molecule tasks searches for molecules that are simultaneously similar to a pair of molecules. This previously has been studied by Brown et al. [41].

- **Multi property optimization.** The {osimertinib, fexofenadine, ranolazine, perindopril, amlodipine, CNS, Pioglitazone} multi property optimization (MPO) tasks attempt to fine-tune the structural and physicochemical properties of known drug molecules. For example, for the sitagliptin MPO benchmark, the models must generate molecules that are as dissimilar to sitagliptin as possible, while keeping some of its properties.

- **Other tasks.** The valsartan SMARTS benchmark targets molecules containing a SMARTS pattern related to valsartan while being characterized by physicochemical properties corresponding to the sitagliptin molecule. Next, the scaffold Hop and decorator Hop benchmarks aim to maximize the similarity to a SMILES strings, while keeping or excluding specific SMARTS patterns, mimicking the tasks of changing the scaffold of a compound while keeping specific substituents, and keeping a scaffold fixed while changing the substitution pattern. The LogP tasks aim at generating molecules with the targeted value of octanol-water partition coefficient. The TPSA task attempts to find a molecule with the targeted value of topological polar surface area. The QED task aims at maximizing the quantitative estimate of drug-likeness score.

# F  Discussion on GuacaMol molecules

| Albuterol similarity | | Ranolazine MPO | |
|:---:|:---:|:---:|:---:|
| **passed** | **rejected** | **passed** | **rejected** |
| 1.000 | 1.000 | 0.921 | 0.946 |
| **Sitagliptin MPO** | | **Zaleplon MPO** | |
| **passed** | **rejected** | **passed** | **rejected** |
| 0.902 | 0.922 | 0.756 | 0.756 |

Figure 10: Illustration of the molecules generated for the GuacaMol benchmark, that have passed and rejected from the filtering procedure. Below each molecule, we also denote the associated objectives.

$\qquad$ (a) $\qquad\qquad\qquad\qquad$ (b) $\qquad\qquad\qquad\qquad$ (c)

Figure 11: (a) The main fragments in the passed molecule of Ranolazine MPO are marked. Two main fragments exist in reality: (b) N-(2-fluoro-6-methylphenyl) formamide, (c) 2-(2-methoxyphenoxy) ethanol.

In this section, we further provide descriptions on the molecules generated from GEGL for the tasks of GuacaMol benchmark, illustrated in Figure 10.

**Molecules passed from the post-hoc filtering.** We first report some of the discovered molecules that pass the post-hoc filtering procedure and have realistic yet novel structures. For example, one may argue that the passed molecule for the Ranolazine MPO (in Figure 10) is chemically realistic; its main fragments of N-(2-fluoro-6-methylphenyl) formamide, (Figure 11b) and 2-(2-methoxyphenoxy) ethanol (Figure 11c) actually exist in reality. We have also verified the molecule to be novel from the chemical databases such as Scifinder [72] and Drugbank [73]. We provide the detailed illustration for the discovered molecule in Figure 11.

**Molecules rejected from the post-hoc filtering.** In Figure 10, we illustrate an example of rejected molecules for the Albuterol similarity, Ranolazine MPO, Sitagliptin MPO, and Zaleplon MPO, respectively. They are filtered out as they are chemically reactive with high probability. Specifically, the molecule from the Albuterol similarity was rejected due to containing a SMILES of `SH`, i.e., *I5 Thiols*. The molecule from the Ranolazine MPO was due to `C=CC=C`[6], and that from the Sitagliptin MPO was due to `NNC=O`, i.e., *acylhydrazide*. Functional groups like `SH`, `C=CC=C`, and `NNC=O` are known to be reactive with high probability, hence they are rejected from the filtering procedure.

## Footnotes

[6]Note that `C=CC=C` in a ring is stable, but `C=CC=C` not in a ring is reactive, i.e., the left side of the molecule.