[Reviews · NeurIPS 2020]

Review 1

Summary and Contributions: The paper combines a neural-based generative model with a genetic algorithm for molecular property optimization. Through a series of experiments, the paper demonstrates significant improvement of the model over the previous state-of-the-art.

Strengths: The paper introduces a novel model and provides a thorough experimental evaluation on standard benchmarks for molecular property optimization. The proposed approach is novel and intuitive—distribution learning and molecular property optimization models cooperate to discover novel structures with the desired properties. Ablation studies demonstrate the improved performance of the full model compared to evaluating separate parts independently.

Weaknesses: 1. While empirically GEGL performs better than the expert alone, the apprentice's contribution is not clear. Since the expert can produce any molecule using atom-wise mutations, it should (in theory) discover the same molecules as GEGL if given enough time. 2. The model without apprentice freezes priority queue Q, as stated in line 257. Line 8 in Algorithm 1 suggests that the expert only modifies molecules from Q and not from Q_{ex}. Hence, the ablated model does not learn from the best molecules discovered in previous iterations and continues mutating and crossing molecules from ChEMBL.

Correctness: The authors use a non-standard implementation of PenalizedLogP. While the authors discuss the difference between two implementations and provide the results on both in supplementary materials, I encourage the authors to use the results from the original implementation (cycle_basis) in the main text.

Clarity: The paper is well written and presented. The authors provide clear motivation behind the design of each experiment. One typo: ChEMBLE→ChEMBL (main text line 258, supplementary line 547)

Relation to Prior Work: The paper provides an overview of many previous models for molecular property optimization. The authors group the previous models into three categories: deep reinforcement learning, deep embedding optimization, and genetic algorithms; the resulting model is the first combination of deep embedding optimization and genetic algorithm.

Reproducibility: Yes

Additional Feedback: Comments: 1. PenalizedLogP was first proposed in "Grammar variational autoencoder" paper, not "Automatic chemical design using a data-driven continuous representation of molecules". The latter optimized non-penalized LogP. 2. Models that optimize PenalizedLogP commonly present an extended list of the best molecules and their scores in supplementary materials. 3. The paper would benefit from additional intuition on why combining apprentice and expert models improves over a single expert model. Questions: 1. Is the set of 800 molecules for constrained optimization of PenalizedLogP the same for both implementations of PenalizedLogP? 800 molecules are usually obtained as molecules with the lowest PenalizedLogP. > UPDATE: the authors said that the set of initial 800 molecules differs from the commonly used set. I suggest the authors provide additional results on the standard set in the camera-ready version. 2. The neural-based generative model is biased towards small molecules, while the genetic algorithm is not. Is GEGL biased towards smaller molecules? How many atoms does the sulfur chain in Table 1 have? What is PenalizedLogP for a molecule `Chem.MolFromSmiles('S'*81)`? > UPDATE: the authors clarified that GEGL is not biased towards smaller molecules 3. What will the result in Figure 6 be if the expert used molecules from Q_{ex} instead of molecules from Q for the genetic algorithm in the expert-only model? Such setup is similar to ChemGE, but since the implementations of genetic algorithms are different, it is necessary to include such a baseline. > UPDATE: the authors provided an additional experiment in their response and demonstrated that GEGL performs better than the genetic algorithm alone. I suggest the authors add such a baseline to other experiments, including logP and all GuacaMol benchmarks to the camera-ready version. UPDATE: The authors answered my questions; I remain positive about the paper and continue to recommend acceptance.


Review 2

Summary and Contributions: This work combines reinforcement learning with genetic algorithms (GA) for generation of optimal molecules for a task. The key contribution is to propose a GA-based refining actor that improves the molecules coming from the generator, and those outcomes act as further training data to improve the generator.

Strengths: The idea is clearly conveyed and direct, and connects with a long track record in chemical design. Chemists have known how to take steps in chemical space for a long time, and expressing the RL challenge around GA in chemistry is refreshing and timely. To some degree even surprising that it had not been done before. Unlike most papers in the field, the authors use a set of benchmark tasks (beyond the infamous LogP). This is commendable, because it allows a fair comparison. From such comparison it is both clear that this approach will produce “good” molecules and also that the challenge is too easy. It’s on the community to propose new metrics that actually represent the challenge of producing molecules.

Weaknesses: One of the strengths of generative algorithms for molecules is to capture a hard-to-describe statistical distribution of plausible molecules that can be made, paid for, stored in a vial, etc. There are many graphs that are formally valid according to valence rules (and their rdkit implementation) that could not exist as molecules because they are not stable. The premise of using generative models for molecular design is sampling natural-looking molecules. Just like generative models for faces, one just needs to look at this to judge whether the model has learned a richer chemistry than the hard-coded rules of RDKit. Very few molecules are shown from what the model produces. It would be great to add samples to the SI and to the accompanying codebase. Like its more naïve predecessor, genetic algorithms, reinforcement learning takes out the guardrails and is likely to generate molecules well out of sample for the training distribution. It is of extra importance for these approaches to describe whether the generated molecules “make sense” and why or why not. There is a reason why genetic algorithms are not really used for chemical design in actual application. After accumulating two or three mutations, the molecules become insanely complicated, unrealistic and out-of-sample with respect to the chemistry that can be actually made. This paper is a great example because their approach works so well. The logP molecules are non-sensical. A linear chain of sulfur atoms (tipped with fluoride?) is the answer a cheeky high school student would find to this problem. Part of this is because the community is asking the wrong question with this LogP task. It’s a meaningless task that a GA in the 1950’s could also solve. But it’s also proof that the model hasn’t learned almost any chemistry, at most, what RDkit will let through or not as valid. The second example for similarity-constrained logP literally took out two charges (and shuffled a bond or two). This is absolutely trivial. The method itself is pretty intuitive and not really a big intellectual or technical jump.

Correctness: Minor issues There are many duplicate references. SI ln 639 “isometric” this word choice seems incorrect. These are isomers. There’s not such thing as isometric molecules. “Hence, DNN-based de novo molecular design might cause safety issues from unexpected behaviors and these aspect should be taken into consideration” The authors underestimate the layers of careful inspection that occur between a neural network generating molecules at machine speed and clinical validation;

Clarity: Yes. Very accessible to lay person. The approach is both simple and clearly explained. No large mathematical explanation needed.

Relation to Prior Work: Yes

Reproducibility: Yes

Additional Feedback: The authors' feedback addressess some of the points above, but the fundamental challenges to the approach remain, in this reviewer's opinion and the score was not changed.


Review 3

Summary and Contributions: This paper proposes to guide deep generative models with genetic algorithms (GA) for molecule optimization. The neural molecule generator is updated to fit the molecules sampled from both the GA algorithm as well as the neural model itself. The method is evaluated on various molecule optimization benchmarks.

Strengths: == Empirical evaluation == The method is evaluated on three molecule generation benchmarks and compared against a comprehensive list of baselines ranging from VAE to deep RL models. The empirical improvement looks significant in all the benchmark datasets. == Reproducibility == The code is attached and all the results are averaged across several runs with standard deviation reported.

Weaknesses: == Generated compounds == For the penalized logP experiment, I am quite concerned about the structures GEGL discovered. The molecule is a chain of sulfurs which I don't feel any chemists would think makes sense. RDKit's logP calculation is actually a regression model. I am afraid that the regression model cannot evaluate logP of this compound (sulfur chains) correctly. Indeed, the benchmark assumes the property calculator (oracle) is perfect, but this assumption barely holds in practice. I am curious how to restrict GEGL to the set of realistic compounds rather than allowing it to discover weird compounds. == Novelty == The proposed method is quite straightforward: it combines genetic algorithm and neural models. The novelty of the technical approach is a bit limited in my opinion. == Lack of discussion to related work == The related work section can be improved a lot. It mentioned several prior works that used genetic algorithms to improve molecule generation, but the technical difference between is not elaborated clearly. For example, what's the difference between GEGL and DA-GA (Nigam et al., 2020)? I had to revisit Nigam et al.'s paper to figure out the exact difference. The difference betwen GEGL and GB-GA should also be clarified.

Correctness: The claims are supported by the experiments and ablation studies. Empirical evaluation follows protocols on standard benchmarks and looks correct to me.

Clarity: The paper is written clearly. I can follow most of the details.

Relation to Prior Work: The related work section can be improved a lot. It mentioned several prior works that used genetic algorithms to improve molecule generation, but the technical difference between is not elaborated clearly. Please see my discussion in the weakness section.

Reproducibility: Yes

Additional Feedback: Upon reading authors feedback, I will keep my original review score.

[Author Response · NeurIPS 2020]

We sincerely thank all reviewers for their valuable efforts and insightful comments. As the reviewers have pointed out,
we believe that our Genetic Expert-Guided Learning (GEGL) framework provides a substantial contribution to the field
with a novel or timely idea (R1, R2), clear writing (All), and extensive evaluations (All). In the following, we provide
our responses to the comments.

———————————————————————— **Response to R1** ————————————————————————

**Unclear contribution of the apprentice policy.** We thank R1 for the helpful comment.
*The apprentice policy contributes to GEGL by encoding knowledge over many molecules*
*seen throughout the training.* This is in contrast to the genetic expert policy which only
use molecules in the priority queue $\mathcal{Q}_{\mathrm{ex}}$ to generate molecules. Especially, the genetic
expert policy alone cannot outperform GEGL since it is likely to meet a poor local
optimum when important "seed" molecules are discarded from the priority queue $\mathcal{Q}_{\mathrm{ex}}$.

Following R1's insightful suggestion, we compared GEGL with an additional "ablation"
algorithm in the right figure. The algorithm is similar to GEGL without the apprentice
policy (in Section 4.3), except the expert policy using molecules from $\mathcal{Q}_{\mathrm{ex}}$ (instead of
$\mathcal{Q}$). We will incorporate this in the final draft, for further clarifying the contribution
of the apprentice policy in GEGL.

**Clarification on details.** We thank R1 for the opportunity to make the following clarifications. First, we indeed used a
different set of low-scoring molecules under different `PenalizedLogP` metrics. Next, we observe that GEGL is not
biased towards generating small molecules; our second-best molecule for optimizing `PenalizedLogP` is a chain of 81
sulfur atoms with `PenalizedLogP` value of 31.790.

——————————————————— **Response to R2 and R3** ———————————————————

**Generated molecules being unrealistic.** We thank R2 and R3 for mentioning an important point. We agree with
R2's comment: the current literature fails to search for a molecule that is high-scoring and realistic simultaneously.[1]
However, we are believe *GEGL can generate high-scoring and realistic molecules under proper regularization, as*
*supported by Table 2(b).* In the experiment for Table 2(b), we apply a post-hoc filter [Brown et al., 2019] for rejecting
unrealistic molecules as suggested by Gao and Coley [2020], and show that GEGL significantly outperforms the
baselines for finding high-scoring molecules even after rejecting many unrealistic molecules. A similar approach can
be used for settings where the oracle score function is unknown (as described by R3), e.g., one may use a DNN that
estimates the true score, while also accounting for the uncertainty of its estimation and realistic-ness of the molecule for
regularization.

Irrespective of the "unrealistic molecule" issue, *the impressive capability of GEGL for finding deficiencies in the scoring*
*functions can be useful* in its own way. To be specific, it is valuable to have methods that can quickly find the limitations
and pitfalls of optimization tasks. Such methods allow us to gain intuition on the problem and to develop better and
rational candidates for the optimal solutions. For example, practitioners have reported many cases for finding bugs of
hardware or simulation while running evolutionary algorithms. We also refer to more detailed discussion on this point
by Lehman et al. [2020].

**Simple method that lacks novelty.** We do believe that our work is novel; GEGL is the first to offer a new paradigm of
combining deep reinforcement learning with domain-specific exploration. Since such a paradigm is not known in the
current literature, it may inspire researchers to develop similar algorithms in other domains. Furthermore, we believe the
simplicity of GEGL is its strength rather than a weakness. Namely, we believe GEGL to be robust, easy to implement,
reproducible, and extendable to broader applications.

——————————— **Response to R1, R2, and R3 (for editorial comments)** ———————————

We plan to fully incorporate the incredibly helpful comments in the final draft, with the following highlights:
**(R1)** We will change Table 2 using the standard `PenalizedLogP` metric, as reported in the supplementary material.
**(R1, R2)** We will report more of the generated molecules in our final draft and the codebase.
**(R3)** We will clarify how DA-GA and GB-GA are different from GEGL; DA-GA only uses a DNN to augment its score
function and GB-GA use the same genetic operator as GEGL without using a DNN.

——————————————————————————— **References** ———————————————————————————

N. Brown et al. Guacamol: benchmarking models for de novo molecular design. *JCIM*, 2019.
W. Gao and C. W. Coley. The synthesizability of molecules proposed by generative models. *JCIM*, 2020.

## Footnotes

[1]As we discuss in Section 4.1, this problem arises for methods regardless of the choice on using a DNN or a genetic algorithm.


[Meta-Review · NeurIPS 2020]

Thank you for your submission. The paper was borderline. R2 main issue was that molecules are not looking realistic. This ties back to observations in https://arxiv.org/pdf/2002.07007.pdf. I fully agree with R2 that we need better models for generating ‘naturally-looking’ molecules. However, I also believe we need better benchmarks for de novo models, and it is not fair to demand from every paper in the literature to invent its own way of evaluating models against this (very important!) objective. The authors also empirically demonstrated the approach can be steered towards more realistic molecules by a proper regularization. To address R2 remark better, I propose that the authors also evaluate how easy it is to synthesize proposed molecules (there are available public APIs, and you could also use ASKCOS). I think that would be much better than the proposed somewhat ad-hoc filtering procedure. All reviewers found the evaluation thorough, and on this grounds I am happy to accept the paper. It presents a simple idea that I believe has chances to nudge the community towards better exploration methods in de novo generation. One issue that came up is that the solution is a bit poorly motivated because genetic operators are hand-crafted. A more perhaps natural rules could be based on feasible chemical reactions as in Molecule Chef. Please include a discussion of such methods in the related work section. Please also remember to address all remarks by the reviewers.